# Synthesis and Characterization of Porous CaCO_3_ Vaterite Particles by Simple Solution Method

**DOI:** 10.3390/ma14164425

**Published:** 2021-08-07

**Authors:** Renny Febrida, Arief Cahyanto, Ellyza Herda, Vanitha Muthukanan, Nina Djustiana, Ferry Faizal, Camellia Panatarani, I Made Joni

**Affiliations:** 1Biotechnology Department, Post Graduate School, Universitas Padjadjaran, Jalan Dipati Ukur No. 35, Bandung 40132, Indonesia; renny.febrida@fkg.unpad.ac.id; 2Functional Nano Powder University Center of Excellence, Universitas Padjadjaran, Jalan Raya Bandung-Sumedang KM 21, Jatinangor, Sumedang 45363, Indonesia; arief.cahyanto@fkg.unpad.ac.id (A.C.); vanithachel@gmail.com (V.M.); nina.djustiana@fkg.unpad.ac.id (N.D.); ferry@phys.unpad.ac.id (F.F.); c.panatarani@phys.unpad.ac.id (C.P.); 3Department of Dental Materials Science and Technology, Faculty of Dentistry, Universitas Padjadjaran, Jalan Raya Bandung-Sumedang KM 21, Jatinangor, Sumedang 45363, Indonesia; 4Department of Dental Materials, Faculty of Dentistry, Universitas Indonesia, Jakarta 10430, Indonesia; ellyza.herda@ui.ac.id; 5Department of Physics, Faculty of Mathematics and Natural Sciences, Universitas Padjadjaran, Jalan Raya Bandung-Sumedang KM 21, Jatinangor, Sumedang 45363, Indonesia

**Keywords:** vaterite, simple solution, spherulitic process, microsphere, mesocrystalline

## Abstract

Appropriately engineered CaCO_3_ vaterite has interesting properties such as biodegradability, large surface area, and unique physical and chemical properties that allow a variety of uses in medical applications, mainly in dental material as the scaffold. In this paper, we report the synthesis of vaterite from Ca(NO_3_)_2_·4H_2_O without porogen to obtain a highly pure and porous microsphere for raw material of calcium phosphate as the scaffold in our future development. CaCO_3_ properties were investigated at two different temperatures (20 and 27 °C) and stirring speeds (800 and 1000 rpm) and at various reaction times (5, 10, 15, 30, and 60 min). The as-prepared porous CaCO_3_ powders were characterized by FTIR, XRD, SEM, TEM, and BET methods. The results showed that vaterite with purity 95.3%, crystallite size 23.91 nm, and porous microsphere with lowest pore diameter 3.5578 nm was obtained at reaction time 30 min, temperature reaction 20 °C, and stirring speed 800 rpm. It was emphasized that a more spherical microsphere with a smaller size and nanostructure contained multiple primary nanoparticles received at a lower stirring speed (800 rpm) at the reaction time of 30 min. One of the outstanding results of this study is the formation of the porous vaterite microsphere with a pore size of ~3.55 nm without any additional porogen or template by using a simple mixing method.

## 1. Introduction

Calcium carbonate is one of the most commonly used compounds in nature and industry because of its biocompatibility and non-toxic properties, as well as other functional structures, and it can fulfill various biological applications such as drug delivery and bone regeneration in dental material as scaffold [1,2,3]. Under ambient conditions, the anhydrous crystalline calcium carbonate forms calcite; otherwise, under certain conditions, it forms aragonite and vaterite. Their crystallization in the natural system is often preceded by the formation and subsequent transformation of amorphous calcium carbonate (ACC) [4,5,6,7,8,9]. Calcite and aragonite are stable forms with trigonal and orthorhombic polymorphic crystals, while vaterite has a hexagonal crystal system. Among other polymorphs, vaterite is the most unstable phase [2,7,8,10,11,12]. Vaterite is used for biomaterial applications such as abrasive agents, bone substitutes, and drug delivery systems [7]. In bone substitute development, porous structure and pure vaterite are preferable to enhance the formation of carbonate apatite [13]. Due to its metastable properties and applications, the synthesis method for preparing pure vaterite and porous structure remains a challenge.

The vaterite particles can be prepared by the solution method with precipitation or carbonation process through CO_2_ bubbling. In general, factors such as solvent, temperature, stirring speed, pH of the medium, ion concentration, and additives influence the morphology or size of the resulting vaterite particles [11,12,14]. For example, the effect of stirring speed was found that a higher agitation speed favored the formation of the smaller size distribution of vaterite microspheres [15]. However, most of these are laborious or require complicated conditions and specialized equipment [6,16]. The simplified method involves mixing saturated aqueous solutions containing calcium and carbonate ions. The mixing method is very cost-effective, quick, and easy to perform and also requires simple instruments. In addition, it can be extended for the industrial production of vaterite particles [7,11,17,18]. The excellent properties of vaterite include a high specific surface area, a higher solubility than calcite and aragonite, a high dispersion, a lower specific gravity, a spherical shape, and a porous internal structure with a particle diameter of 0.05 to 5 μm [11,19,20]. However, controlling the phase transformation of ACC to form pure vaterite is still a challenge. Therefore, there are two main tasks in the synthesis of porous vaterite CaCO_3_: to create the transformation of ACC to pure vaterite and formation of a porous structure.

Among the numerous factors that influence the precipitation of calcium carbonate polymorphs, one of the most determining factors is the presence of various foreign ions or molecules in the aqueous solution from which the carbonate precipitates. Controlling phase transformation of ACC to form pure vaterite governed by their kinetic of ions and molecules (from precursor or additive), and the saturation index of the solution resulted in either remain vaterite or aragonite or calcite. Those factors are determined by the reaction temperature, aging times, reaction times, and stirring speed [8,20]. Jiang (2018) reported that increasing aging times 0 to 42 h and reaction temperatures from 0 °C to 60 °C, correspondingly, reduced the amount of vaterite from 90.4% to 81.4% and 85.8% to 70.2% [20]. Ševčík et al. (2015) found the optimal synthesis conditions for the preparation of pure vaterite (≥99 wt.%) at 60 °C and a stirring speed of 600 rpm without additives using CaCl_2_·H_2_O and KCO_3_ as precursors [8]. It is also remarked that the phase transformation of ACC to calcite and vaterite occurred at a short reaction time of 3 min and eventually at particular conditions either remain calcite or transform into vaterite [10]. Due to the unstable phase of vaterite, other researchers proposed an additional ethylene glycol (EG) or biopolymers, such as carboxymethyl inulin (CMI), served as a stabilizer to prevent the conversion of vaterite to other CaCO_3_ polymorphs (aragonite or calcite) in the dynamic of precipitation process [21,22]. In contrast, fast stirring may also inhibit the conversion of vaterite to calcite [15]. Furthermore, stirring speed affects the size and morphology where higher stirring speeds favored the formation of smaller size and size distribution microspheres. In the formation of porous structures, the use of porogen as a template is usually introduced in the synthesis [4,11,23]. There are very few studies on the synthesis of mesoporous carbonate vaterite using only deionized water as a solvent for a particular precursor without any addictive substances or porogen. To this point, the synthesis of pure and porous vaterite in aqueous solutions is still a major challenge since the products usually have a non-uniform shape, and also calcite and aragonite co-exist.

Although many researchers reported the successful synthesis of pure vaterite and porous structure, the properties of vaterite prepared from different calcium sources resulted in different morphologies and mechanical properties [24]. The mechanical properties of vaterite obtained from Ca(NO_3_)_2_·4H_2_O precursor receive higher shear strength than CaCl_2_ [24]. The application of vaterite for biocompatible ceramic needs to have characteristics similar to mineral phases in the natural bone, such as transformation to phosphate base biomaterial which can be used as bone substitutes and osteoconductive scaffold. The formation of calcium phosphate has received great attention due to its requirement for excellent biocompatibility and osteoconduction as a bone substitute material. Ideal calcium phosphate for bone substitutes shall support cell attachment, migration, proliferation, interact actively with cells and tissues, and stimulate repair and regeneration provided Ca/P ratio closed to 1.67 [25]. There are two recognized techniques for the synthesis of calcium phosphate, either using the solution method or sintering the vaterite in the presence of the phosphate source (DCPA). Many phases of calcium phosphate might form, such as hydroxyapatite, carbonate apatite, and α or ß-TCP indicated by their calcium/phosphorus ratio (Ca/P). If calcium phosphate is prepared by solution method with calcium nitrate as the source, the hydroxyapatite resulted in Ca/P near 1.5 [26] and created biphasic nature. In our future scenario, we prepared the calcium phosphate using the second technique by introducing a phosphor source in vaterite powder during the sintering process. Therefore, an appropriate structure of powder solid particles is required for this process and the Ca(NO_3_)_2_·4H_2_O is a preferable choice for the synthesis of vaterite.

Therefore, the main objective of this study is to synthesize vaterite from Ca(NO_3_)_2_·4H_2_O without porogen to obtain a pure and porous microsphere as appropriate raw material for calcium phosphate. Nature and crystallization behavior of the solids and the evolution of the aqueous solution chemistry were investigated at various reaction times (5, 10, 15, 30, and 60 min) at two selected temperatures (20 and 27°C) and stirring speeds (800 and 1000 rpm) considering optimum condition as reported in the literature [8]. The selected experimental condition can avoid complicated procedural steps, for example, keeping the temperature under room temperature and higher stirring speed. Furthermore, this study provided crucial insight into the calcium carbonate polymorphic transformation processes and porous formation that occurred in the presence of ions or molecules from selected Na_2_CO_3_ precursor and Ca(NO_3_)_2_·4H_2_O precursor in an aqueous environment.

## 2. Materials and Methods

Two precursor materials were used: Na_2_CO_3_ (Natrium Carbonate) and Ca(NO_3_)_2_·4H_2_O (Calcium Nitrate Tetrahydrate); both sources are analytical grade (Merck). The synthesized vaterite was prepared by mixing 0.5 M Na_2_CO_3_ with 0.5 M Ca(NO_3_)_2_·4H_2_O, as illustrated in Figure 1. The solution was continuously mixed under stirring at the speed of 800 and 1000 rpm. As the precipitation process occurs at temperatures of 20–40 °C, reactions were conducted at two conditions of temperatures (20 and 27 °C) for different reaction times (5, 10, 15, 30, and 60 min). The sample was collected and filtered on Whatman filter paper in a Buchner funnel at various reaction times. The obtained precipitate was washed with ethanol several times to ensure the removal of mother liquor. The washed precipitate was then dried in a desiccator for 48 h.

The synthesis of vaterite was performed at two different temperatures and two different stirring speeds. The samples were investigated as the reaction times increased. Samples were measured in transmission mode by FTIR Nicolet iS5 (Thermo Scientific, Waltham, MA, USA) equipped with the iD5 ATR. FTIR analysis was performed to investigate the absorption bands of CaCO_3_ polymorphs. The obtained spectra exhibit the characteristic absorption bands that correspond to symmetric stretching, out-of-plane bending, asymmetric stretching, and in-plane bending [27]. The spectral region was selected from 1000 cm^−1^ to 600 cm^−1^ for the analysis. 

The crystal structure of synthesized vaterite was analyzed by X-ray diffraction (XRD) (PANalytical, Almelo, Netherlands). Diffraction data were acquired by exposing samples to Cu-Kα X-ray radiation, which has a characteristic wavelength of 1.5406 Å. The X-rays were generated from a Cu anode using accelerating voltage and the applied current 40 kV and 30 mA, respectively. The data were collected within the range of 20 to 60° and scan step time 2.90 s. The corresponding XRD patterns aimed to confirm the presence of both calcite and vaterite structures. The crystallite size of the microsphere was quantitatively calculated based on the Debye–Scherrer Equation to investigate the effect of the reaction time on the formation of vaterite microspheres, as follows Equation (1):(1)$$D=\frac{k\lambda }{\beta cos\theta }$$
where *D* is the size of the crystal, k is the Debye–Scherrer constant (0.89), λ is the wavelength of X-ray (1.5406 Å), β is the line broadening from the full width at half maximum (FWHM), and θ is the Bragg angle. To know the percentage of occurred crystal phase in the samples, we conducted quantitative analysis on the XRD spectra using MATCH (Crystal Impact, Bonn, Germany, version 3.7.0.124), which can perform a semiquantitative analysis of the sample using the so-called Reference Intensity Ratio Method [28].

The morphology of the particle was examined by scanning electron microscope (SEM) (Hitachi SU3500, Tokyo, Japan) with an accelerating voltage of 10 kV. A light source was introduced into the cell, and the scattered light was collected at 90°. The size distribution of primary and secondary particles was obtained using image analysis software ImageJ (NIH Image, Bethesda, MD, USA, version 1.46r: Java 1.6.0_20) from the magnification of SEM images. High-resolution transmission electron microscopy was conducted on Hitachi TEM System with an accelerated voltage of 120 kV.

The specific surface area was measured by the Brunauer–Emmett–Teller (BET) method using Quantachrome instruments surface area analyzer (Quantachrome Instrument, Boynton Beach, FL, USA). The small number of samples are dried with nitrogen purging (vacuum) that applies high temperature with pressure tolerance (*P*/*P*_0_) of 0.050/0.050 (ads/des) as a measurement point. The outgas temperature was 300 °C. The gas volume adsorbed to the surface of the samples is measured at the nitrogen boiling point. It was correlated to the total surface area of the samples, including pore volume and pore diameter, which was calculated based on the BET Equations (2)–(6) [29]:(2)1W((P0P)−1)=1WmC+C−1WmC (PP0)
where,
W = weight of gas adsorbed$\frac{P}{P_{0}}$ = relative pressure$W_{m}$ = weight of adsorbate as monolayerC = BET constant

After conducting the multipoint BET method, three data points are shown, i.e., volume at STP, relative pressure $\left(\frac{P}{P_{0}}\right)$ as x-axis, and 1W((P0P)−1) as y-axis in the multipoint BET linear plot. The intercept (i) is related to the first term of Equation (2) as follows:(3)$$i=\frac{1}{W_{m}C}$$

Meanwhile, the slope (s) is related to the part of the second term of equation (2) as follows:(4)$$s=\frac{C-1}{W_{m}C}$$

The total surface area ($S_{t}$) can be obtained as follows:(5)$$S_{t}=\frac{W_{m}\;N\;A_{cs}}{M}$$
where,
$W_{m}$ = weight of adsorbate as monolayerN = Avogadro’s number (6.023 × 10^23^)M = Molecular weight of adsorbate$A_{cs}$ = Adsorbate cross sectional area (16.2 for Å^2^ Nitrogen)

The specific surface area (S) is then determined by dividing the total surface area ($S_{t}$)

with sample weight (w):(6)$$S=\frac{S_{t}}{w}$$

## 3. Results and Discussion

### 3.1. The Characteristics of Porous CaCO_3_ Microsphere

Figure 2 and Figure 3 show FTIR and XRD spectra of the prepared carbonate polymorphs as a function of reaction time where their absorption is indicated by V (vaterite) and C (calcite) at temperature 20 °C and different stirring speeds (800 and 1000 rpm). Fourier Transform Infrared spectroscopy was used to track changes in carbonate-related vibrational modes in three different CaCO_3_ polymorphs (calcite, aragonite, and vaterite). One carbonate ion can have four normal modes: symmetric stretching, out-of-plane bending, asymmetric stretching, and in-plane bending [8,27]. The results showed the characteristic absorption band of vaterite at 849 cm^−1^, 877 cm^−1^, and 744 cm^−1^ and absorption bands of calcite at 877 cm^−1^ and 712 cm^−1^. Overlapping of vaterite and calcite absorption bands were present at 877 cm^−1^.

Considering only characteristics of active carbonate modes, the in-plane bending mode shows the most pronounced changes as a function of reaction time at a particular temperature and stirring speed. We have shown that spontaneous precipitation of calcium carbonates from moderately or low supersaturated (S ≤ 1) solutions led to the initial formation of vaterite and calcite. The FTIR spectra in Figure 2a show that lower intensity of the calcite absorption band is obtained at reaction time from 5 to 15 min. These results indicated that at this condition (temperature 20 °C and stirring speed 800 rpm), crystals mainly consist of vaterite and calcite present as a minor phase. However, the presence of calcite was increased at the reaction duration of 30 and 60 min. This result is evidence of precipitate instability of ACC formation to form the vaterite phase at lower temperatures and lower stirring speed. The stirring speed created hydrodynamic conditions of ions and molecules, promoting collisions or diffusion of the vaterite crystal with intense contact with the solution. Therefore, much amount of vaterite transformed into calcite upon increasing their reaction time (30–60 min). However, the kinetics of the process was also to be dependent on the ionic strength of the solution. Three possible kink sites occur to be a calcium site ≡CO_3_Ca^+^, a carbonate site ≡CaCO_3_^−^ or a bicarbonate site ≡CaHCO_3_°. A step at the calcite surface advances or retreats through the addition or loss of calcium and (bi-) carbonate ions at growth sites. To maintain crystal stoichiometry, the net rates at which calcium and (bi-)carbonate ions are incorporated at kink sites must be equal.

The corresponding XRD pattern also confirmed that the obtained CaCO_3_ polymorphs were a mixture of calcite and vaterite (Figure 2b). The characteristic peaks and hkl plane (Table 1) of calcite were detected at 29.32°, 35.90°, 39.36°, 47.47°, 48.43°, and 57.37°; meanwhile, peaks of vaterite were detected at 20.89°, 24.82°, 26.99°, 32.72°, 43.71°, 49.07°, 50.06°, and 55.74°. Like the FTIR observation, high vaterite with the lowest calcite peaks was observed for the reaction time of 15 min. The calculated percentage of calcite and vaterite phases of CaCO_3_ polymorphs were shown in Table 2. A high percentage vaterite phase (97.1%) was obtained or present in the mesocrystalline with the lowest crystallite size (23.90 nm). Thus, both XRD and FTIR analysis showed that at the reaction time of 15 min received the highest vaterite phase and calcite present as a minor phase and also a relatively high percentage of vaterite (95.3%) at reaction time 30 min with crystallite sized 23.91 nm. A higher percentage of calcite (37.8%) in reaction times of 10 min compared to 30 min (4.7%) can be accepted due to its higher peak intensity of 877 cm^−1^ in the FTIR result, which confirmed its presence in the higher calcite phase. Unlike the stirring speed 800 rpm, upon the increase of stirring speed (1000 rpm), the FTIR spectra (Figure 3a) showed the highest calcite absorption band at 712 cm^−1^ indicated the highest percentage of calcite phase in the CaCO_3_ polymorphs for the reaction time 15 min. This analysis is also supported by the XRD observation, where their corresponding higher calcite peaks were obtained at 15 min (Table 3). The higher stirring speed creates higher hydrodynamic conditions of ions and molecules improved possibility of collisions of the vaterite crystal with intense contact with the solution caused the enhanced amount of vaterite transformed into calcite. Generally, the result on varying the stirring speed at temperature 20 °C appeared to vaterite as the major component of CaCO_3_ polymorphs, but still unstable phase due to moderate supersaturation at low temperature (20 °C).

The sample at reaction time 30 min was further investigated by SEM and TEM due to its relatively small crystallite size compared to others. The spherical morphology of CaCO_3_ was obtained for the sample at the reaction time of 30 min at two different stirring speeds as shown in Figure 4a–d. The CaCO_3_ particles formed secondary microspheres with nanosize primary spherical particles. The size distribution analysis showed that the average size of microsphere was about (2.91 ± 1.06) μm for the stirring speed of 800 rpm (Figure 4a), and about (2.47 ± 0.85) μm for the stirring speed of 1000 rpm (Figure 4c). The average size of the primary particles were (153 ± 27.95) nm (Figure 4b) and (171.29 ± 36.61) nm (Figure 4d) at corresponding stirring speed 800 and 1000 rpm. For the stirring speed 800 rpm, the morphology was denser and more spherical compared to the morphology at stirring speed of 1000 rpm, relatively irregular in shape. This was due to the higher percentage of the vaterite phase in the sample (up to 95.3%). At the same reaction time, the average size of the microsphere was higher at stirring speed 800 compared to 1000 rpm. This means that the growth rate of the microsphere is faster at lower speeds. Thus, it was emphasized that at higher stirring speed received smaller microsphere particles due to higher speed may prevent adsorption of the nanoparticle suspension in the process of spherulitic growth to form larger incorporation of nanoparticles in the presence of centrifugal force. However, when the stirring speed increased, the motion of ions and molecules improved caused increasing the chances of collision and increasing the rate of vaterite active interaction with foreign ions in the solution promoted phase transformation of vaterite to calcite as shown in SEM images Figure 4c which is highlighted in the circle. This picture is in agreement with the previous FTIR and XRD analysis in Figure 2 and Figure 3. Generally, all obtained sample at 20 °C undergoes phase transformation of vaterite metastable state. Thus, further study was extended for the higher temperature (27 °C) to see the phase transformation, morphology, and porosity of the obtained porous particles and investigate their formation mechanism.

In contrast to the FTIR and XRD analysis of the sample obtained at 20 °C, lower calcite phase formation as a function of reaction times at temperature 27 °C as shown in Figure 5 and Figure 6. It was highlighted that at stirring speed 800 rpm, the calcite phase indicated in peak absorption was reduced significantly as a function of times. However, at the beginning of the reaction, calcite formation dominates over vaterite (reaction time ≤ 5 min) at a lower stirring speed (800 rpm). It was highlighted that at the temperature of 27 °C appeared to vaterite as the major component of CaCO_3_ polymorphs, with metastable phase due to higher supersaturation compared to temperature 20 °C.

The XRD pattern of polymorphs was shown in Figure 5b, along with their attribution of the main peaks. From the XRD patterns, the characteristic peaks corresponding to vaterite were dominant as the reaction time increased, which was confirming that a higher temperature of 27 °C can also inhibit the transformation of vaterite to calcite. The detailed quantitative results based on XRD analysis were listed in Table 4 and Table 5. According to the calculation, the crystallite sizes of microspheres were calculated as 29.94, 26.25, 26.00, 23.92, and 25.08 nm, respectively (Table 4). This showed that the crystallite size tends to decrease with the increase of reaction times from 5 to 60 min (at the temperature of 27 °C and stirring speed of 800 rpm). The peaks corresponding to calcite were much narrower than those of vaterite, confirming a larger crystallite size.

The XRD result confirming the FTIR spectroscopy, when the sample was stirred under 800 rpm at 5 min, only calcite was obtained, and the crystallite size was higher (29.94 nm). When the reaction time increased, the crystallite size was decreased to 25.08 nm as the percentage of calcite was decreased (Table 4). It was seen from the XRD pattern, that vaterite percentage was increased and much metastable vaterite crystal was obtained as the increase of reaction time. When the samples were mixed under a stirring speed of 1000 rpm, the calculated crystallite sizes were 25.11, 22.99, 24.33, 22.58, and 23.90 nm, respectively (Table 5). This showed that the crystallite size decrease with the increase of reaction time from 5 to 60 min (at the temperature of 27 °C and stirring speed of 1000 rpm). However, based on the calculated mean of crystallite size showed all value was not significant and in agreement with result previously reported research [30]. This means the selected experimental condition revealed the crystallite size relatively homogeneous and also similar morphology observed in the SEM images.

SEM images of the samples mixed at 27 °C under stirring speeds of 800 rpm and 1000 rpm for 30 min reaction time were shown in Figure 7a–d along with their 5 and 20k magnification. It was observed from the size distribution that microspheres average size were (2.6 ± 0.9) μm (Figure 7a) and (2.94 ± 0.99) μm (Figure 7c) corresponding for stirring speed 800 and 1000 rpm. These microspheres consisted of primary nanosize particles with average size (107.59 ± 21.76) nm and (177.07 ± 41.18) nm respectively for stirring speed 800 rpm (Figure 7b) and 1000 rpm (Figure 7d). The only small amount of calcite was present in the sample and relatively spherical. More spherical vaterite was obtained under stirring speed of 800 rpm compared with stirring speed 1000 rpm. TEM analysis also confirmed the morphology of obtained vaterite microstructure, as shown in Figure 8a–d. The TEM images revealed the CaCO_3_ microsphere consisted of many agglomerated primary nanosize particles with relatively homogeneous in size around 150 nm.

To analyze the total surface area of vaterite particles, multipoint BET measurement was conducted, as shown in the second column of Table 6. The results showed that the specific surface area is greatly affected by the stirring speed. The specific surface decreases when the stirring speed increased. The BJH (Barret, Joyner, and Halneda) method was conducted to calculate the pore surface area, volume, and diameter from experimental isotherm using adsorption and desorption technique. Only BJH desorption results are shown in the table. The result showed that only stirring speed affects the surface area of the CaCO_3_ polymorphs. The pore surface area, volume, and diameter remain the same with the increase of temperature. This indicated that the temperature difference between 20 °C and 27 °C did not affect their surface area. Meanwhile, when the stirring speed increased, the surface area and pore diameter were decreased, but the pore diameter was increased. This might be due to the diffusion of NaNO_3_ from the solution to the microsphere trapped during the spherulitic process.

### 3.2. The Proposed Mechanism of Spherulitic Growth of CaCO_3_ Mesoporous

The mechanism of the ACC to vaterite formation is still under development, revealed with three main proposed mechanisms or models. The first proposed mechanism starts by dissolving ACC to form spherical vaterite by homogeneous nucleation of crystalline vaterite nanoparticles, and consequently, vaterite nanoparticles aggregate very rapidly to form the polycrystalline microsphere [31,32,33]. Second, ACC particles are dehydrated and recrystallized to form vaterite [34,35]. Third, continuous dissolution of ACC and spherulitic growth formation of vaterite microsphere [36,37,38,39,40]. Therefore, in this study, we investigated the effect of temperature and stirring speed as a function of reaction times on the phase transformation vaterite crystals, morphology, and also polycrystalline pore structure. It was explained by considering the mechanism of the vaterite crystallization by introducing a discussion on the recent proposed model crystal growth mechanism. One of the outstanding results of this study was the formation of porous particles without any additional porogen or template and interestingly using a simple mixing method. The kinetic formation of vaterite in the presence of various ions and molecules represents an intermediate step in the reaction pathway that leads from ACC to vaterite following Ostwald’s steps rule [36]. The most important one was the mechanism on the formation of the porous formation of polycrystalline microsphere without additional porogen or template. The formation of porous polycrystalline vaterite was markedly different from the conventional particle to particle formation in which nucleation occurred.

The process involved mixing a precursor of Na_2_CO_3_ as the source of CO_3_ and Ca(NO_3_)_2_·4H_2_O source of Ca prepared at the condition of supersaturation concerning the initial precursors results in the formation of amorphous calcium carbonate (ACC) which later transform into vaterite crystal. Many factors influence the precipitation of calcium carbonate polymorphs, one of the most determinant factors was the presence of foreign ions (Ca^2+^, Na^+^, CO_3_^2−^) or molecules (NaNO_3_) in the aqueous solution from which the carbonate precipitates. The proposed chemical reaction for the used precursors Na_2_CO_3_ and Ca(NO_3_)_2_·4H_2_O are presented in Equations (1)–(3) and excess of NaNO_3_ molecules in the solution (Equation (2)). The complete chemical reaction was presented in Equation (3).
Ca(NO_3_)_2_ 4H_2_O (s) → Ca(NO_3_)_2_ (aq) + 4H_2_O(7)
Na_2_CO_3_ (aq) + Ca(NO_3_)_2_ (aq) → 2NaNO_3_(aq) + CaCO_3_(s)(8)
Na_2_CO_3_(aq) + Ca(NO_3_)_2_ 4H_2_O (aq)→ 2NaNO_3_(aq) + CaCO_3_(s) + 4H_2_O(9)

We proposed the mechanism of polycrystalline vaterite microsphere formation at temperature 27 °C based on the FTIR, XRD, SEM, and TEM observations, as illustrated in Figure 9. Immediately after mixing the precursors, nucleation of Ca^2+^ and CO_3_^2−^ ions occurred spontaneously in supersaturation solution to form clustered of primary vaterite crystals transformation to growth. The vaterite crystals are fully formed at near-equilibrium saturation conditions. On the other hand, Ostwald ripening of the small nanocrystallites is formed as the result of the internal crystal structure changing over time. Smaller nanocrystals divorcement due to a total contact area reduction with a solvent causes the growth of larger ones. That formation above occurred continuously with the fast spherulitic growth of vaterite crystal to form a mesocrystalline vaterite microsphere via growth front nucleation (GFN) mechanism [38,39].

The primary vaterite crystals have a secondary structure called polycrystalline, which is composed of nanocrystallites that have a size in the range of nanometers [41]. Na^+^ and NO_3_^−^ ions could be adsorbed in the original primary vaterite nanosize particles surface inside the microsphere and consequently prevent the crystal growth polymorph and phase transformation. A similar mechanism was reported using Na_2_CO_3_ as a carbon source that ions such as NH_4_^+^ ions are able to control the growth of primary particles [42]. In this case, the particle growth mechanism of spherulites of vaterite was found to be dependent on the crystal surface structure. The GFN mechanism mostly produced polycrystalline sphere contained crystalline nanosphere with similar in size. This is in agreement with the morphology observed in SEM and TEM images (Figure 7 and Figure 8). The presence of excess NaNO_3_ molecules in the solution may be diffused inside of the microsphere during spherulitic growth and has a significant role for the mesocrystalline particles. In this condition, the trapped NaNO_3_ diffusion prevented primary particles from becoming new crystals in secondary size during the formation of spherulites of the vaterite microsphere [41]. Thus, pore mesocrystalline microsphere was obtained with relatively similar pore size which was consistent with the observation of their morphology in SEM and TEM images (Figure 7 and Figure 8) and BET of pore size in Table 6. Thus, the presence of the NaNO_3_ acting as porogen in microsphere during spherulitic growth was able to control the crystal porosity without the addition of any chemicals. Therefore, the obtained microspheres represent a very important biomaterial for various biomedical application such as bone substitute, i.e., our future development of CaCO_3_ vaterite based scaffold.

## 4. Conclusions

In conclusion, we have successfully synthesized vaterite from Ca(NO_3_)_2_·4H_2_O without porogen to obtain vaterite with purity 95% crystallite size 23.91 nm and porous microsphere with lowest pore diameter 3.5578 nm at reaction time 30 min, temperature reaction 20 °C, and stirring speed 800 rpm. It was emphasized that more spherical with a size of around 2–3 µm and consisted of multiple primary nanoparticles to form the porous microsphere with lower pore size at the lower stirring speed (800 rpm) at the reaction time of 30 min. Generally, the percentage of vaterite and calcite co-exist were varied as the reaction time increase in all temperatures and stirring speed. It was concluded that the BET result confirmed that only stirring speed affects the surface and pore volume, and consequently pore diameter due to NaNO_3_ act as porogen in the spherulitic process of mesocrystalline vaterite microsphere. Considering the result over the ranges of variables (temperature and stirring speed), the experimental route presented in this paper offering the efficient procedure to obtain a high percentage yield of porous vaterite (majority more than 90%). Therefore, it is potentially feasible for developing into industrial scale production.

## Figures and Tables

**Figure 1 materials-14-04425-f001:**
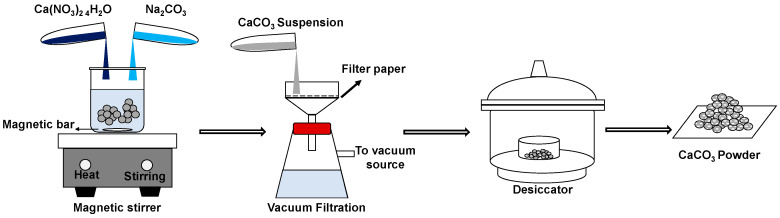
Schematic illustration of synthesis CaCO_3_ by a simple procedure.

**Figure 2 materials-14-04425-f002:**
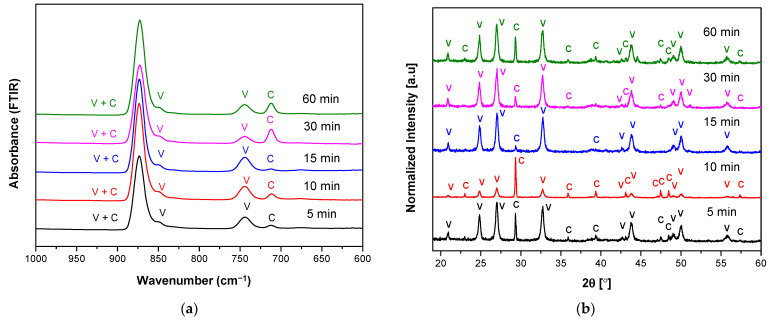
(**a**) FTIR and (**b**) XRD spectra of vaterite synthesized at 20 °C for different reaction times (5–60 min) at stirring speed 800 rpm. Polymorphs main peak is indicated (V = vaterite, C = calcite).

**Figure 3 materials-14-04425-f003:**
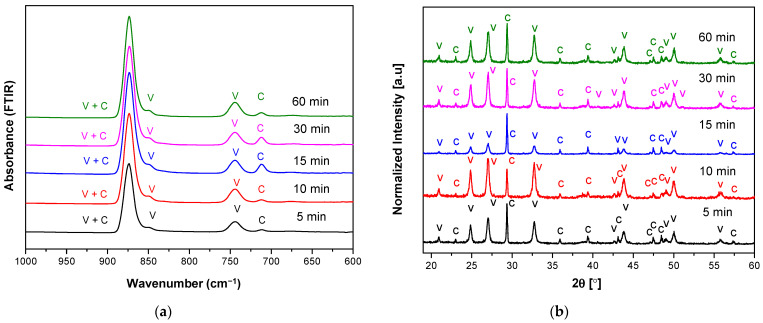
(**a**) FTIR and (**b**) XRD spectra of vaterite synthesized at 20 °C for different reaction times (5–60 min) at stirring speed 1000 rpm. Polymorphs main peak is indicated (V = vaterite, C = calcite).

**Figure 4 materials-14-04425-f004:**
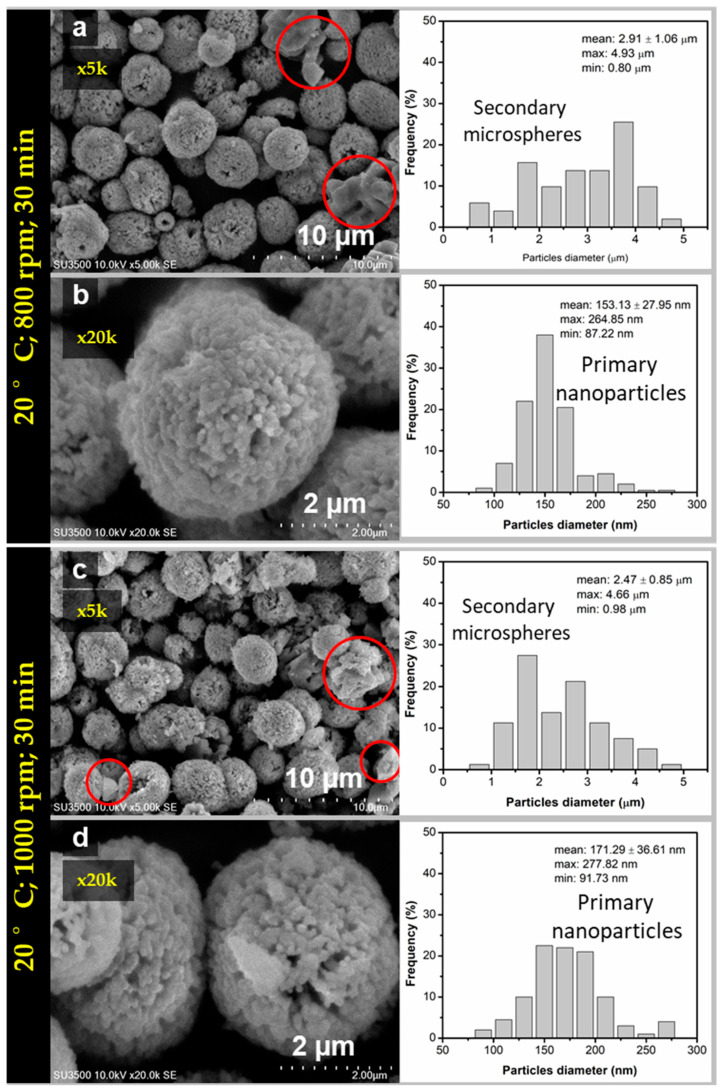
SEM images and particles size distribution of vaterite synthesized at 20 °C (time reaction 30 min) for different stirring speeds (**a**,**b**) 800 rpm and (**c**,**d**) 1000 rpm.

**Figure 5 materials-14-04425-f005:**
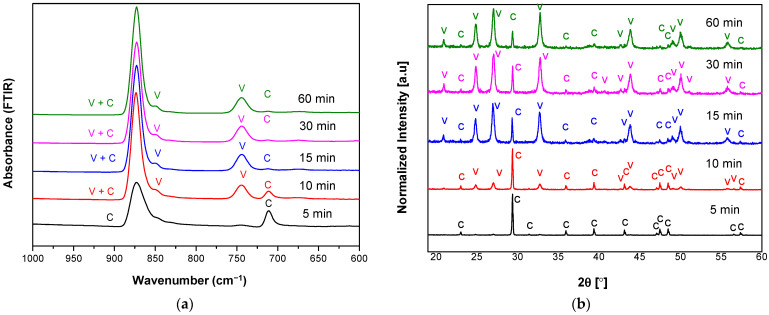
(**a**) FTIR and (**b**) XRD spectra of vaterite synthesized at 27 °C for different reaction times (5–60 min) at stirring speed 800 rpm. Polymorphs main peak is indicated (V = vaterite, C = calcite).

**Figure 6 materials-14-04425-f006:**
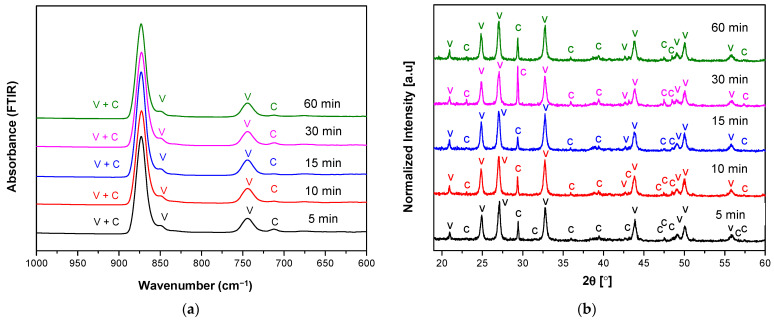
(**a**) FTIR and (**b**) XRD spectra of vaterite synthesized at 27 °C for different reaction times (5–60 min) at stirring speed 1000 rpm. Polymorphs main peak is indicated (V = vaterite, C = calcite).

**Figure 7 materials-14-04425-f007:**
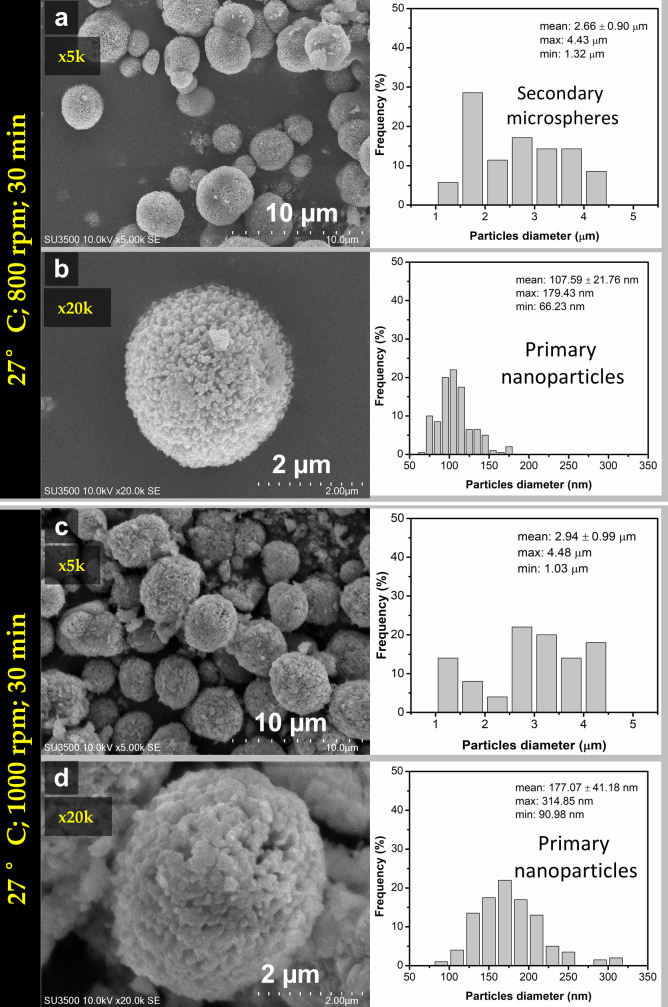
SEM images and particles size distribution of vaterite synthesized at 27 °C (time reaction 30 min) for different stirring speeds (**a**,**b**) 800 rpm and (**c**,**d**) 1000 rpm.

**Figure 8 materials-14-04425-f008:**
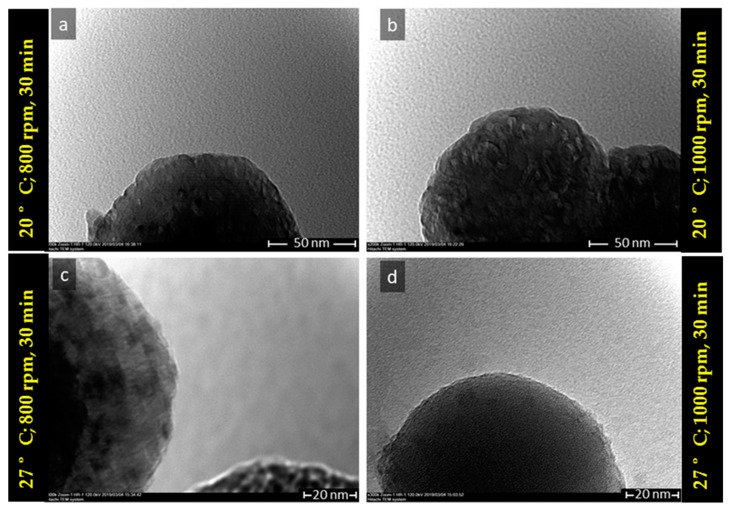
TEM images of vaterite synthesized at (**a**) T = 20 °C, 800 rpm, (**b**) T = 20 °C, 1000 rpm, (**c**) T = 27 °C, 800 rpm, and (**d**) T = 27 °C, 1000 rpm (reaction time 30 min).

**Figure 9 materials-14-04425-f009:**
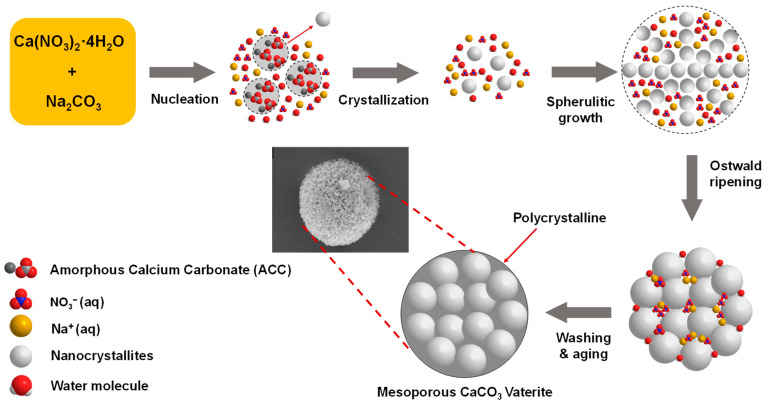
The illustration of the proposed mechanism of spherulitic growth of porous CaCO_3_ Microsphere.

**Table 1 materials-14-04425-t001:** hkl plane of vaterite and calcite peaks.

No.	Vaterite		Calcite	
	hkl	2Theta [deg]	hkl	2Theta [deg]
1	002	20.89	104	29.32
2	100	24.82	110	35.90
3	101	26.99	113	39.36
4	102	32.72	018	47.47
5	110	43.71	116	48.43
6	112	49.07	122	57.37
7	104	50.06	-	-
8	202	55.74	-	-

**Table 2 materials-14-04425-t002:** Effect of reaction time on the as-prepared CaCO_3_ polymorph ratio and size of crystallite (at a temperature of 20 °C and stirring speed of 800 rpm).

Sample Name	Reaction Time (min)	Calcite (%)	Vaterite (%)	Crystallite Size (nm)
R05 T20 S800	5	8.2	91.8	26.02
R10 T20 S800	10	37.8	62.2	27.18
R15 T20 S800	15	2.9	97.1	23.90
R30 T20 S800	30	4.7	95.3	23.91
R60 T20 S800	60	12.7	87.3	24.83
				Mean = 25.17 ± 1.42

**Table 3 materials-14-04425-t003:** Effect of reaction time on the as-prepared CaCO_3_ polymorph ratio and size of crystallite (at a temperature of 20 °C and stirring speed of 1000 rpm).

Sample Name	Reaction Time (min)	Calcite (%)	Vaterite (%)	Crystallite Size (nm)
R05 T20 S1000	5	19.0	81.0	24.16
R10 T20 S1000	10	13.5	86.5	27.82
R15 T20 S1000	15	36.3	63.7	25.07
R30 T20 S1000	30	17.8	82.2	26.34
R60 T20 S1000	60	14.4	85.6	27.57
				Mean = 26.19 ± 1.58

**Table 4 materials-14-04425-t004:** Effect of reaction time on the as-prepared CaCO_3_ polymorph ratio and size of crystallite (at a temperature of 27 °C and stirring speed of 800 rpm).

Sample Name	Reaction Time (min)	Calcite (%)	Vaterite (%)	Crystallite Size (nm)
R05 T27 S800	5	100	0	29.94
R10 T27 S800	10	46.3	53.7	26.25
R15 T27 S800	15	8.1	91.9	26.00
R30 T27 S800	30	9.6	90.4	23.92
R60 T27 S800	60	5.9	94.1	25.08
				Mean = 26.24 ± 2.26

**Table 5 materials-14-04425-t005:** Effect of reaction time on the as-prepared CaCO_3_ polymorph ratio and size of crystallite (at a temperature of 27 °C and stirring speed of 1000 rpm).

Sample Name	Reaction Time (min)	Calcite (%)	Vaterite (%)	Crystallite Size (nm)
R05 T27 S1000	5	7.5	92.5	25.11
R10 T27 S1000	10	5.6	94.4	22.99
R15 T27 S1000	15	4.3	95.7	24.33
R30 T27 S1000	30	6.5	93.5	22.58
R60 T27 S1000	60	6.8	93.2	23.90
				Mean = 23.78 ± 1.02

**Table 6 materials-14-04425-t006:** Effect of temperature (20 °C, 27 °C) and stirring speed (800 rpm, 1000 rpm) on the specific surface area measured by the Brunauer–Emmett–Teller (BET) method (reaction time 30 min).

Sample	Specific Surface Area (m^2^/g)	Pore Surface Area (m^2^/g)	Pore Volume (cc/g)	Pore Diameter Dv [d] (nm)
T20 SS800	10.853	20.640	0.036	3.5578
T20 SS1000	4.767	13.093	0.030	3.5731
T27 SS800	10.331	20.715	0.042	3.5757
T27 SS1000	7.721	13.724	0.026	3.5801

## Data Availability

The data set generated and analyzed during the study are available upon reasonable request to the author.

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
