# Peer review of "Synthesis and Characterization of Porous CaCO3 Vaterite Particles by Simple Solution Method"

_materials, 2021, doi:10.3390/ma14164425_

Round 1

Reviewer 1 Report

Materials 1309075

Title: Synthesis and Characterization of Porous CaCO3 Vaterite Particles by Simple Solution Method

Authors: R. Febrida, A. Cahyanto, E. Herda, V. Muthukanan, N. Djustiana, F. Faizal, C. Panatarani, I.M. Joni

General Comments:

The manuscript by Febrida et al., reports on the synthesis and characterization of porous calcium carbonate particles with emphasis on the vaterite polymorph. A very simple solution based synthesis method was employed and the effect of temperature and stir rate are studied as it pertains to the evolution of calcite and vaterite phases as a function of reaction time.

The manuscript is carefully written and organized but does suffer from issues with sentence construction and grammar. These issues affect clarity of the discussion. In terms of technical content, the detail of the proposed mechanism does not appear to be supported by sufficient evidence. Most importantly, the article claims to be entirely focused on synthesis of vaterite, but all conditions studied have significant amounts of calcite preset. In addition, most of the claims are based on very small differences (e.g., crystallite sizes of 4 nm calculated by the Debye-Scherrer equation). This may stem from the relatively small variations in the experimental conditions (e.g., 20 vs. 27 °C; 800 vs. 1000 rpm).

I do not recommend publishing the article currently.  Further experiments, analysis, and editing for English grammar are all needed. I suggest the authors resubmit after extensive revisions.  Some comments are included below to help the authors in this process.

Specific Comments: (in no particular order)

  1. The authors place great emphasis on the use of porous vaterite in biomedical applications. How the reported vaterite may be useful for such applications is never discussed after the introduction. Perform additional experiments that demonstrate utility of the synthesized vaterite in such applications (e.g., solubility, mechanical properties).
  2. Only two points for each experimental variable are probed. Two temps and two stir speeds isn’t enough data to illustrate the dependence of polymorph formation and porosity on temperature and speed. Perform experiments at a minimum of two stir speeds and temperatures, carefully designing the experiments so that a span of at least an order of magnitude in the value of each variable is probed.
  3. The Debye Scherrer equation has a great deal of limitations, which impart high uncertainty to the predicted values. The small differences in crystallite size reported in the manuscript may not actually be significant.  Line broadening in pXRD peaks isn’t exclusively a representation of particle size.  See:  Kim Y-Y., Schenk A. S., Ihli J., Kulak A. N., Hetherington N. B. J., Tang C. C., Schmahl W. W., Griesshaber E., Hyett G. and Meldrum F. C. (2014) Nature Commun. “A Critical Analysis of Calcium Carbonate Mesocrystals", 5, 4341 DOI:10.1038/ncomms5341
  4. Include a table with the hkl plans for each peak in the pXRD patterns.
  5. Line 254 mentions a bimodal distribution of spheres: Based on what? Was image analysis performed?  If not, then please do so and show the histogram for the size distribution, based on an n value of at least 100.
  6. Figures with microscopy content would be easier to interpret if the experimental conditions were labeled in the body of the figures. For example, in Figs. 7&8, arrange the quad of images so that there is an additional column with the value of the experimental variable(s) listed.
  7. The mechanism proposed in Figure 9 isn’t justified by the data. Additional experiments are required.  Many researchers have worked on this very topic.  For example, see the works of B. Pokroy, L. Benning, H. Cölfen, and others.

Reviewer 2 Report

I have to say my overall impressions are positive. The authors provided here a detailed characterization of the exciting and essential compound. One can find information about the structure, morphology, or size of obtained particles.  Also, the language and the style are acceptable. In my opinion, the presented results are of good quality and well described; therefore, the paper deserves to be considered for publication. Also, all tables, figures, and formulas are provided with good quality. I have no comments here. The authors are asked to address the comments below:

  1. The equations should be numbered consequently through the whole manuscript. Therefore the equations 1, 2, and 3 in the lines 363-365 should be re-numbered to 7, 8 and 9
  2. The authors provided a detailed characterization with many parameters determination. But there is no word or even comment about the uncertainty of the obtained results. Many parameters were calculated according to given equations, where some variables were derived from the experiments. In this way always some uncertainty should be taken into account, or if it is small or negligible, it should be commented.

In summary, I recommend the minor revision of the paper, especially the authors are asked to provide some uncertainty analysis.

Author Response

General Comments:

I have to say my overall impressions are positive. The authors provided here a detailed characterization of the exciting and essential compound. One can find information about the structure, morphology, or size of obtained particles.  Also, the language and the style are acceptable. In my opinion, the presented results are of good quality and well described; therefore, the paper deserves to be considered for publication. Also, all tables, figures, and formulas are provided with good quality. I have no comments here. The authors are asked to address the comments below:

Response for reviewer general comments:

We acknowledge reviewer for the valuable suggestions to improve the quality of this article. We have provided responds for comments and questions raised by reviewer. Additional explanations are also added into the manuscript to adopt the reviewer suggestion.

Specific Comments:

Point 1:
The equations should be numbered consequently through the whole manuscript. Therefore, the equations 1, 2, and 3 in the lines 363-365 should be re-numbered to 7, 8 and 9

Response 1:       

Thank you for the correction. We have revised the paragraph of the equations 1, 2, and 3 (lines 374-376)  by re-numbering the equations to be 7, 8 and 9 in the manuscript.

Point 2:
The authors provided a detailed characterization with many parameters determination. But there is no word or even comment about the uncertainty of the obtained results. Many parameters were calculated according to given equations, where some variables were derived from the experiments. In this way always some uncertainty should be taken into account, or if it is small or negligible, it should be commented.

Response 2:                                                                                                         

We agree with your comment about the uncertainty of the obtained results, i.e., in the calculated crystallite size from Debye-Scherrer Equation. Therefore, we added the mean value and standard deviation from reaction times for each table of crystallite size (Table 2, 3, 4, 5) and explained about the uncertainty of the obtained results in discussion section (line 324-327) as follows:

“However, based on the calculated mean of crystallite size showed all value was not significant and in agreement with result previously reported research [30]. This means the selected experimental condition revealed the crystallite size relatively homogeneous and similar morphology was also observed in the SEM images.”

Round 2

Reviewer 1 Report

The authors have made improvements to the article through the first round of reviews; however, the impact of this manuscript is still limited by the small range over which the experimental variables were probed.  

Author Response

This manuscript is a resubmission of an earlier submission. The following is a list of the peer review reports and author responses from that submission.